

# Sarcopenia and echocardiographic parameters for prediction of cardiovascular events and mortality in patients undergoing maintenance hemodialysis

Mengyan Zhang[1,*], Liuping Zhang[1,*], Yezi Hu[2], Ying Wang[1], Shengchun Xu[1], Xiaotong Xie[1], Tian Xu[1], Zuolin Li[1], Hui Jin[2] and Hong Liu[1]

[1] Zhongda Hospital, Southeast University School of Medicine, Institute of Nephrology, Nanjing, Jiangsu Province, China
[2] Zhongda Hospital, Southeast University School of Medicine, Institute of Nutrition, Nanjing, Jiangsu Province, China
* These authors contributed equally to this work.

Corresponding author
Hong Liu, jstzliu@sina.com

## ABSTRACT

**Background:** Sarcopenia is prevalent and is associated with the occurrence of cardiovascular complications in patients undergoing maintenance hemodialysis (MHD). It is unknown how skeletal muscle may be associated with aspects of myocardial structure and function. This study aimed to evaluate the association between sarcopenia and cardiac structure and function in patients undergoing MHD. We also examined the prognostic role of sarcopenia for mortality and cardiovascular events (CVE) in this population.

**Methods:** Participants from a single center underwent bioimpedance body composition analysis to measure skeletal muscle and echocardiography to assess myocardial structure and function. Sarcopenia was diagnosed based on the Asian Working Group for Sarcopenia criteria. The end points were all-cause mortality and CVE.

**Results:** Of the 158 participants, 46 (29.1%) had sarcopenia, 102 (64.6%) had left ventricular diastolic dysfunction (LVDD), and 106 (67.0%) had left ventricular hypertrophy (LVH). Participants with sarcopenia had smaller right ventricular sizes (2.54 ± 0.77 *vs* 2.76 ± 0.28; *P* < 0.01), inter-ventricular thickness (1.07 ± 0.19 *vs* 1.14 ± 0.20; *P* = 0.039), and left ventricular posterior wall thickness (0.96, 0.89–1.10 *vs* 1.06, 0.95–1.20; *P* = 0.018). Skeletal muscle mass was strongly correlated with left ventricular mass (LVM) (r = 0.577; *P* < 0.0001). Furthermore, the risk of LVDD (OR: 4.92, 95% confidence interval (CI) [1.73–13.95]) and LVH (OR: 4.88, 95% CI [1.08–21.96]) was much higher in the sarcopenic group than in the non-sarcopenic group. During a follow-up period of 18 months, 11 (6.9%) patients died, of which seven died (4.4%) of CVE, and 36 (22.8%) experienced CVE. The presence of sarcopenia (adjusted hazard ratio (HR), 6.59; 95% CI [1.08–39.91]; *P* = 0.041) and low skeletal muscle index (HR, 3.41; 95% CI [1.01–11.57]; *P* = 0.049) and handgrip strength (HR, 0.88; 95% CI [0.78–0.99]; *P* = 0.037) independently predicted death.

Sarcopenia was a significant predictor of CVE (HR, 10.96; 95% CI [1.14–105.10]; $P = 0.038$).

**Conclusion:** Our findings demonstrated that sarcopenia is associated with LVDD and LVH, and is associated with a higher probability of death and CVE.

## INTRODUCTION

Hemodialysis is the most common form of kidney replacement therapy worldwide, comprising approximately 69% of all renal replacement therapy and 89% of dialysis cases. Recent studies have shown that the number of patients undergoing maintenance hemodialysis (MHD) is increasing annually worldwide (*Thurlow et al., 2021*). Mortality among patients on MHD is significantly higher than that among their counterparts in the general population (*Bello et al., 2022*). Sarcopenia is a common complication in patients undergoing MHD and is considered an important predictor of low quality of life, cardiovascular disease (CVD), and mortality (*Lai et al., 2019*).

Sarcopenia is a progressive and generalised skeletal muscle disorder involving the accelerated loss of muscle mass and function that is associated with increased adverse outcomes including falls, functional decline, frailty, and mortality (*Cruz-Jentoft & Sayer, 2019*). The overall prevalence of sarcopenia in community-dwelling older adults aged over 65 years in the Chinese population ($n = 25,921$) was 17.4% (*Ren et al., 2022*). Patients undergoing MHD are more prone to sarcopenia due to risk factors such as increased nutrient losses in dialysate, sedentary lifestyle leading to inactivity and a chronic inflammatory state (*Noce et al., 2021*). According to a meta-analysis of pooled data, the prevalence of sarcopenia in patients undergoing MHD was 31% (*Shu et al., 2022*).

Recent studies have shown that changes in skeletal muscle mass (SMM) are closely related to CVD, and sarcopenia can increase the incidence of cardiovascular events (CVE) and lead to a poor prognosis (*Lena, Anker & Springer, 2020*). Several observational studies have assessed the relationship between decreased muscle mass and cardiac function and CVD, and found that patients with sarcopenia were more likely to develop CVD, such as myocardial infarction, angina pectoris, and congestive heart failure (*Veronese et al., 2017*).

Cardiac structure and function change was associated with a higher risk of CVD. It is unknown how skeletal muscle may be associated with aspects of myocardial structure and function. The newly discovered cardio-muscular axis relationship confirms that sarcopenia is related to a reduction in the size of the left ventricle and atrium, and that SMM is independently correlated with indicators of myocardial structure (*Keng et al., 2019*). Moreover, low skeletal muscle mass induce endothelial inflammation and insulin resistance and eventually alter myocardial structure and function, which may progress (*Yoo et al., 2021*). In a retrospective study of patients with type 2 diabetes, sarcopenia was associated with a high risk of LVDD and LV remodeling (*Zhang et al., 2021*). Due to its economical, simple, and effective advantages in measuring myocardial quality,

echocardiography provides a convenient and reliable method for clinical diagnosis of cardiac structure and function (*Beyer et al., 2018*).

If cardiac structural and functional dysfunction have associated with sarcopenia, these might help to identify of sarcopenic persons who are at especially high risk for the CVD events. Therefore, we investigated to analyze the correlation between sarcopenia and cardiac structure and function evaluated by several echocardiographic indices, to evaluate the impact of sarcopenia on mortality and CVE in patients undergoing MHD. A better understanding about the association between sarcopenia and cardiac structure and function can contribute to advances in developing prognostic models and new therapeutic targets.

## MATERIALS AND METHODS

### Study design and population

This was a prospective, observational study. Patients undergoing MHD at Zhongda Hospital, Southeast University, China, were recruited from December 2020 to May 2022. The inclusion criteria were (1) age >18 years, (2) MHD for 4 h three times per week for at least 3 months, and (3) clinical stability (defined as no hospitalization required within 3 months). The exclusion criteria were: (1) difficulty in maintaining a standing position or completing the six-meter (6-m) walk test; (2) presence of artificial implants, such as cardiac pacemakers or artificial joints; (3) physical disability; (4) severe cognitive disorder and mental illness; and (5) severe pleural effusion and/or ascites. The study was approved by the Independent Ethics Committee (IEC) for Clinical Research of Zhongda Hospital, affiliated with Southeast University (batch number: 2021ZDSYLL230-P01) and was conducted in compliance with the tenets of the Declaration of Helsinki. All the patients provided written informed consent for participation in the study and data processing.

### Echocardiography

Echocardiography was performed using Vivid 7 (GE Medical Systems, Milwaukee, WI, USA) by trained sonographers and clinicians, using standardized guidelines. Echogenic parameters, including left ventricular ejection fraction (LVEF), transmitral early diastolic velocity (E), and mitral annulus early diastolic velocity ($e'$), were assessed. LVEF was assessed using Simpson's rule *via* manual tracing of biplane digital images. Pulse-wave Doppler transmitral inflow velocity was obtained from an apical five-chamber view to assess diastolic dysfunction. The mitral $E/e'$ ratio was used as an index of the LV diastolic filling pressure. Linear measurements of the left posterior wall thickness (PWT), intraventricular septum thickness (IVST), and diameter of the left ventricular cavity at the end of diastole (LVIDd) and systole (LVIDs) were obtained in the M-mode in the parasternal long-axis view. LV mass (LVM) was calculated based on measurements obtained in M-mode using the following equation: LVM (g) = 0.8 × [1.04 × (LVIDd + IVST + PWT)3 − (LVIDs)3] + 0.6. Left ventricular mass index (LVMI) was calculated as LVM/body surface area (*Soeding et al., 2020*). LVH was defined as an LVM/BSA ratio >95 g/m$^2$ in women and >115 g/m$^2$ in men. The relative wall thickness was calculated as (interventricular septal (SWT) + PWT)/EDD, using a cut-off value of 0.42 to define

eccentric (≤0.42) or concentric (>0.42) remodeling (*Nardi et al., 2021*). LVDD was defined as one of the following: (1) E/e ratio greater than 15, (2) E/A ratio greater than 2, or (3) E/A ratio <1 (*Silbiger, 2019*).

## Measurements of body composition and muscle strength

Bioelectrical impedance analysis (BIA) was conducted to determine the skeletal muscle index (SMI) using a multifrequency BIA device (InBody 770; InBody Co., Ltd. Seoul, Korea), according to the manufacturer's instructions. To increase the accuracy of the results, BIA was performed after adequate dialysis.

Handgrip strength (HGS) was measured for each participant using a Takei handgrip dynamometer (Guangdong Xiangshan Weighing Apparatus Group, Zhongshan, China) following standard protocols. HGS was measured before the initiation of dialysis to avoid the confounding effect of the dialysis process. Participants were instructed to stand upright with their arms at their sides. The handgrip dynamometer was held with the indicator facing outward, and the grip width was adjusted such that the second joint of the pointing finger made a right angle with the dynamometer. The participants were then instructed to grip the instrument with full force. Measurements were performed on the opposite arm of that with the arteriovenous fistula while participants stood with the arms along the body. Measurements were repeated twice and the highest HGS values were recorded.

Measurement of 6-m Gait speed (GS) was performed before the start of dialysis by using a stopwatch. The requirement for the measurement is that the patients need to maintain their usual pace until the end of the measurement. Gait velocity was calculated by dividing the distance walked (*i.e.*, 6 m) by the time it was completed. The test was repeated twice and the average speed was calculated. All measurements of body composition, HGS, and gait speed were performed by the same trained operator.

Sarcopenia was defined based on the criteria outlined by the Asian Working Group for Sarcopenia (*Chen et al., 2020*). For men and women, the cutoff values for HGS and SMI were <28 kg and <7.0 kg/m$^2$, and <18 kg and <5.7 kg/m$^2$, respectively. The cutoff value for GS was <1.0 m/s. And the criteria were based on the presence of low muscle mass index as an essential criterion, accompanied by either low HGS or slow GS.

## Follow-up study

After sarcopenia screening, the patients were followed up at the outpatient clinic for 18 months. The endpoints of this study were CVE (cardiovascular death, stroke, nonfatal myocardial infarction, unstable angina, coronary intervention (coronary artery bypass surgery or angioplasty), heart failure (HF), and peripheral artery disease) and all-cause mortality. Mortality and CVE were identified by referring to medical records and confirmed by direct contact with the patients, relatives, and in-charge physicians.

## Statistical analysis

All statistical analyses were performed using SPSS version 26.0 (SPSS Inc., IBM Corporation, Armonk, NY, USA). Values are presented as mean ± standard deviation or as median and interquartile range, as appropriate, for continuous variables. Categorical

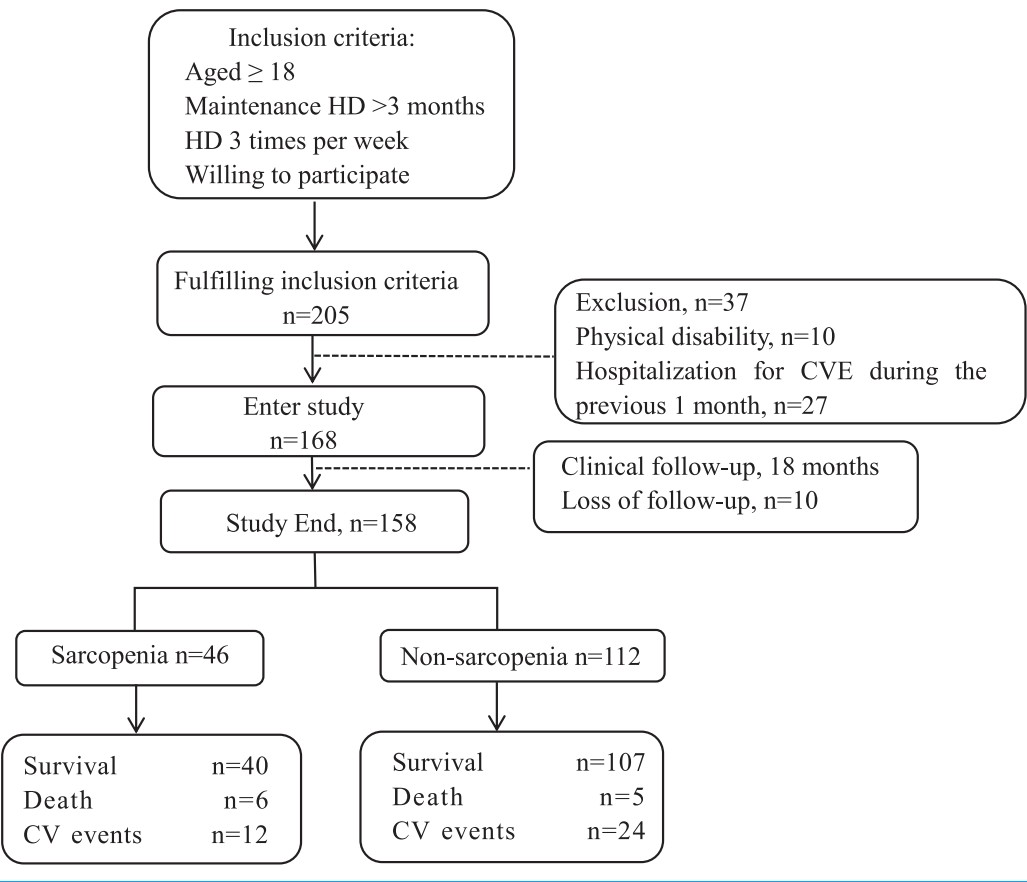

**Figure 1 Enrollment flow chart for this study.** HD, hemodialysis; CVE, cardiovascular events.

variables are presented as numbers and percentages. The Kolmogorov–Smirnov test was used to evaluate the normality of the distribution. Pearson's correlation analysis was used to clarify the relationship between body composition and cardiac parameters. Logistic regression tests were performed to explore the correlations among the assessed parameters. Survival curves were plotted using the Kaplan–Meier method and evaluated using the log-rank test. Hazard ratios (HRs) for mortality were determined *via* both crude and multivariate Cox regression analyses and are presented with 95% confidence intervals (CIs). Statistical significance was set at $P < 0.05$.

## RESULTS

The enrolment flowchart for this study is shown in Fig. 1. The source population of this single-center cohort study included 205 patients who met the inclusion criteria. Among these, 37 patients were excluded, 10 of them due to physical disability and the rest due to hospitalization of CVE within 1 month previously, and 10 patients were lost to follow-up during 18 months. Thus, 158 patients were included in the data analysis, 46 (29.1%) of whom had sarcopenia. The mean follow-up duration was 18 months, during which adverse CVE and all-cause deaths were recorded (shown in Fig. 1).

**Table 1 Baseline characteristics of the study participants.**

| | Non-sarcopenia (n = 112) | Sarcopenia (n = 46) | Total (n = 158) | P-value |
|---|---|---|---|---|
| Age, y | 54 ± 14 | 62 ± 12 | 56 ± 14 | <0.0001 |
| Sex (male, %) | 51 (45.5%) | 27 (58.7%) | 78 (49.4%) | 0.133 |
| Smoking (%) | 94 (83.9%) | 38 (82.6%) | 132 (83.5%) | 0.839 |
| Hypertension (%) | 90 (80.4%) | 37 (80.4%) | 127 (80.4%) | 0.991 |
| Diabetes mellitus, n (%) | 25 (22.3%) | 10 (21.7%) | 35 (22.2%) | 0.936 |
| Prevalent CVD, % | 38 (33.9%) | 26 (56.5%) | 64 (40.5%) | 0.009 |
| Prevalent cerebrovascular disease, % | 12 (10.7%) | 6 (13.2%) | 18 (11.4%) | 0.675 |
| On antihypertensive treatment, % | 48 (42.9%) | 24 (52.2%) | 72 (45.6%) | 0.285 |
| On statins, % | 4 (3.6%) | 1 (2.2%) | 5 (3.2%) | 0.648 |
| Systolic blood pressure, mmHg | 135 (125, 151) | 140 (126, 155) | 137 (127, 152) | 0.697 |
| Diastolic blood pressure, mmHg | 75 (68, 85) | 72 (66, 79) | 75 (67, 85) | 0.111 |
| BMI, kg/m$^2$ | 22.72 (20.55, 25.83) | 20.01 (18.40, 21.93) | 21.93 (19.78, 24.73) | <0.0001 |
| Skeletal muscle mass, kg | 24.60 (21.43, 29.37) | 21.95 (18.25, 24.75) | 23.42 (20.74, 27.85) | 0.003 |
| Skeletal muscle index, kg/m$^2$ | 6.88 ± 1.01 | 5.89 ± 0.87 | 6.59 ± 1.07 | <0.0001 |
| HGS, kg | 25.61 (18.23, 32.35) | 18.20 (13.05, 23.90) | 21.92 (16.55, 29.85) | <0.0001 |
|   Male | 33.34 (27.00, 38.30) | 23.79 (19.40, 27.10) | 30.03 (21.88, 35.18) | <0.0001 |
|   Female | 20.16 (14.60, 25.15) | 13.08 (8.30, 16.70) | 18.48 (12.23, 22.28) | <0.0001 |
| GS, m/s | 0.93 (0.74, 1.05) | 0.79 (0.69, 1.06) | 0.88 (0.73, 1.05) | 0.207 |
| HB, g/L | 106.09 ± 16.66 | 107.19±19.57 | 106.41±17.50 | 0.722 |
| PTH, ng/L | 280.16 (145.87, 539.63) | 205.25 (84.86, 444.45) | 269.90 (128.42, 521.01) | 0.249 |
| Total cholesterol, mg/dl | 3.91 (3.38, 4.45) | 3.90 (3.08, 4.74) | 3.91 (3.38, 4.51) | 0.883 |
| Triglycerides, mg/dl | 1.46 (0.93, 1.15) | 1.49 (1.15, 1.93) | 1.46 (0.99, 2.02) | 0.985 |
| HDL-C, mg/dl | 1.02 (0.88, 1.31) | 1.14 (0.96, 1.36) | 1.05 (0.91, 1.31) | 0.256 |
| LDL-C, mg/dl | 2.13 (1.82, 2.60) | 2.36 (1.52, 2.76) | 2.18 (1.80, 2.75) | 0.803 |
| CRP, mg/dl | 2.63 (0.90, 7.99) | 1.85 (0.82, 8.34) | 2.33 (0.83, 8.14) | 0.635 |
| Albumin, g/L | 41.15 (39.03, 43.05) | 38.90 (36.60, 42.03) | 40.5 (38.2, 42.8) | 0.009 |
| Kt/V | 1.25 (1.24–1.27) | 1.25 (1.23–1.27) | 1.25 (1.24–1.27) | 0.721 |
| Time of dialysis, year | 5.90 (3.30, 9.95) | 6.85 (3.60, 9.78) | 6.05 (3.40, 9.85) | 0.548 |

Notes:
  Data are presented as mean values ± standard deviation, n (%), or median (interquartile range).
  CVD, cardiovascular disease; BMI, body mass index; HB, hemoglobin; PTH, parathyroid hormone; HDL-C, high-density lipoprotein cholesterol; LDL-C, low-density lipoprotein cholesterol; CRP, C-reactive protein; Kt/V, a measure of dialysis adequacy; K, dialyzer clearance of urea; t, dialysis time; V, volume of urea distribution.

Table 1 shows the baseline characteristics according to the sarcopenic status. Compared with participants without sarcopenia, participants with sarcopenia were older ($P < 0.0001$) and had a lower BMI ($P < 0.0001$). Participants with sarcopenia had lower overall SMM ($P = 0.003$), lower SMI ($P < 0.0001$), and lower HGS ($P < 0.0001$). The serum albumin level also gradually decreased in the sarcopenia group ($P < 0.001$).

On echocardiography, 102 participants (64.6%) had LVDD and 106 (67.0%) had LVH. Participants with sarcopenia had lower inter ventricular septum thickness (IVST) ($P = 0.039$) and LVPW (left ventricular posterior wall thickness) values ($P = 0.018$). Participants with sarcopenia also had smaller right ventricular (RV) sizes ($P < 0.01$; Table 2). SMM was strongly and positively correlated with LVM (r = 0.577; $P < 0.0001$; shown in Fig. 2).

**Table 2 Cardiovascular measurements.**

| Echocardiography variables | Non-sarcopenia (n = 112) | Sarcopenia (n = 46) | Total (n = 158) | P-value |
|---|---|---|---|---|
| LVST, mm | 1.14 ± 0.20 | 1.07 ± 0.19 | 1.12 ± 0.20 | 0.039 |
| LVPW, mm | 1.06 (0.95, 1.20) | 0.96 (0.89, 1.10) | 1.03 (0.92, 1.16) | 0.018 |
| LVEDd, mm | 4.92 (4.41, 5.30) | 4.81 (4.40, 5.30) | 4.88 (4.4, 5.3) | 0.377 |
| LVM, g | 201.64 (156.89–248.79) | 180.83 (136.84–220.50) | 194.37 (153.04, 242.17) | 0.071 |
| LVMI, g/m² | 118.25 (97.19, 143.97) | 113.71 (95.52, 145.91) | 116.93 (96.33, 144.42) | 0.481 |
| LVEF (%) | 0.69 (0.64, 0.74) | 0.67 (0.62, 0.71) | 0.68 (0.64, 0.73) | 0.200 |
| RWT, mm | 0.44 ± 0.07 | 0.42 ± 0.09 | 0.43 ± 0.08 | 0.164 |
| RVD, mm | 2.76 ± 0.28 | 2.54 ± 0.77 | 2.49 ± 0.36 | <0.01 |
| RAD, mm | 3.90 (3.49, 4.30) | 3.75 (3.30, 4.10) | 3.80 (3.40, 4.20) | 0.101 |
| E/e′ | 14.00 (10.00, 18.00) | 15.70 (12.00, 19.88) | 14.00 (10.00, 19.00) | 0.579 |
| LVDD (%) | 61 (54.5%) | 41 (89.1%) | 102 (64.6%) | <0.0001 |

Note:
IVST, inter ventricular septum thickness; LVPW, left ventricular posterior wall thickness; LVEDd, left ventricular end-diastole diameter; LVM, left ventricular mass; LVMI, left ventricular mass index; LVEF, left ventricular ejection fraction; RWT, relative wall thickness; RVD, right ventricular diameter; RAD, right atria diameter; E/e′, peak E velocity flow in early-diastole period/peak e velocity flow in end-diastole period; LVDD, left ventricle diastolic dysfunction.

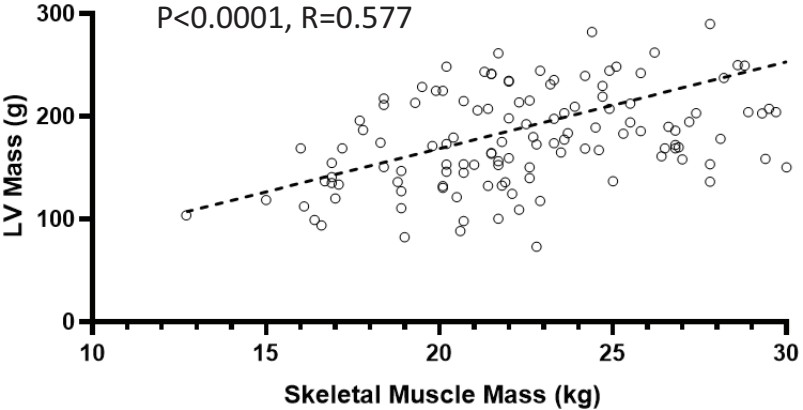

**Figure 2 Relationship between skeletal muscle mass and left ventricular (LV) mass.**

**Table 3 Univariate and multivariate associations of clinical variables with diastolic dysfunction.**

| | Univariate | | Multivariate | |
|---|---|---|---|---|
| | Unadjusted OR (95%) | P-value | Adjusted OR (95%) | P-value |
| Sarcopenia | 6.86 [2.52–18.64] | <0.0001 | 4.92 [1.73–13.95] | 0.003 |
| Age | 1.06 [1.03–1.08] | <0.0001 | 1.05 [1.01–1.08] | 0.01 |

Note:
Age, hypertension, diabetes mellitus, and serum creatinine were included in the multivariate model after adjustment for multiple confounding factors such as age, presence of diabetes, hypertension, and serum creatinine. OR, odds ratio.

Table 3 presents the results of the binary logistic regression analyses that revealed an association between sarcopenia and the prevalence of LVDD. In comparison with the non-sarcopenia group, the sarcopenia group was independently associated with the presence of LVDD, with odds ratio (OR) of 6.86 (95% CI [2.52–18.63]) and, following

**Table 4 Univariate and multivariate associations of clinical variables with left ventricular hypertrophy.**

| | Univariate | | Multivariate | |
|---|---|---|---|---|
| | Unadjusted OR (95%) | *P*-value | Adjusted OR (95%) | *P*-value |
| Hypertension | 2.28 [1.02–5.08] | 0.044 | 2.08 [0.69–6.21] | 0.192 |
| Age, y | 1.02 [0.99–1.04] | 0.135 | 1.03 [0.98–1.08] | 0.172 |
| HGS, kg | 0.98 [0.96–1.01] | 0.256 | 0.95 [0.90–0.99] | 0.020 |
| SMI, kg/m$^2$ | 1.29 [0.93–1.78] | 0.129 | 2.55 [1.24–5.23] | 0.011 |
| GS, m/s | 0.66 [0.17–2.64] | 0.556 | 2.32 [0.17–31.62] | 0.527 |
| Sarcopenia | 0.78 [0.38–1.59] | 0.489 | 4.88 [1.08–21.97] | 0.039 |
| BMI | 1.02 [0.93–1.11] | 0.706 | 0.59 [0.40–0.87] | 0.008 |
| LVDD (%) | 0.83 [0.41–1.68] | 0.613 | 0.59 [0.18–1.89] | 0.375 |

**Note:**

The analysis was adjusted for multiple confounding factors such as age, sex, presence of diabetes, hypertension, diastolic dysfunction, LVEF, BMI, RWT, hemoglobin, skeletal muscle index, handgrip, and walk speed. HGS, handgrip strength; SMI, skeletal muscle index; GS, Gait speed; BMI, body mass index; LVEF, left ventricular ejection fraction; RWT, relative wall thickness; LVDD, left ventricular diastolic dysfunction; OR, odds ratio.

adjustment for multiple confounding factors such as age, presence of diabetes, and hypertension, 4.92 (95% CI [1.73–13.95]).

The association between sarcopenia and the presence of LVH was determined using binary logistic regression analyses (Table 4). In comparison with the non-sarcopenia group, the sarcopenia group was independently associated with the presence of LVH, with an OR of 4.88 (95% CI [1.08–21.97]) following adjustment for multiple confounding factors such as age, presence of diabetes, and hypertension. Higher BMI (adjusted OR: 0.59; 95% CI [0.40–0.87]; *P* = 0.008), lower handgrip strength (adjusted OR: 0.95; 95% CI [0.90–0.99], *P* = 0.020), and smaller LA (adjusted OR: 10.01; 95% CI [3.41–29.33], *P* < 0.0001) were associated with a higher risk of LVH.

Data from 158 patients undergoing MHD were available for the analysis of mortality and CVE. During the follow-up period, 36 (22.8%) patients were hospitalized for CVD and 11 (6.9%) passed away. The following events were registered: five (3.2%) cases of cardiovascular-related death; five (3.2%) cases of death due to infection, and two (1.3%) cases of death from an unknown cause. Six (4.4%) patients experienced nonfatal myocardial infarction, five (3.2%) were hospitalized for stroke; two (1.3%) experienced peripheral artery disease, and 18 (11.4%) had HF. Kaplan–Meier analysis demonstrated a significantly higher probability of death in the sarcopenia group than in the non-sarcopenia group (log-rank test, *P* = 0.045) (shown in Fig. 3A). Patients with sarcopenia had a higher risk of CVE than those without sarcopenia (log-rank test, *P* = 0.05) (shown in Fig. 3B).

Cox proportional hazards analyses (Table 5) indicated a significant mortality risk in groups with SMI (HR, 3.41; 95% CI [1.01–11.57]; *P* = 0.049), older age (HR, 1.10; 95% CI [1.02–1.18]; *P* = 0.041), lower HGS (HR, 0.88; 95% CI [0.78–0.99]; *P* = 0.037), and sarcopenia (HR, 6.59; 95% CI [1.08–39.91]; *P* = 0.041) when adjusting for age, prevalence of CVD, albumin, HGS, BMI, and LVMI. Sarcopenia was a strong predictor of CVE (HR, 10.96; 95% CI [1.14–105.10]; *P* = 0.038) and mortality (HR, 6.59; 95% CI [1.08–39.91]; *p* = 0.041).

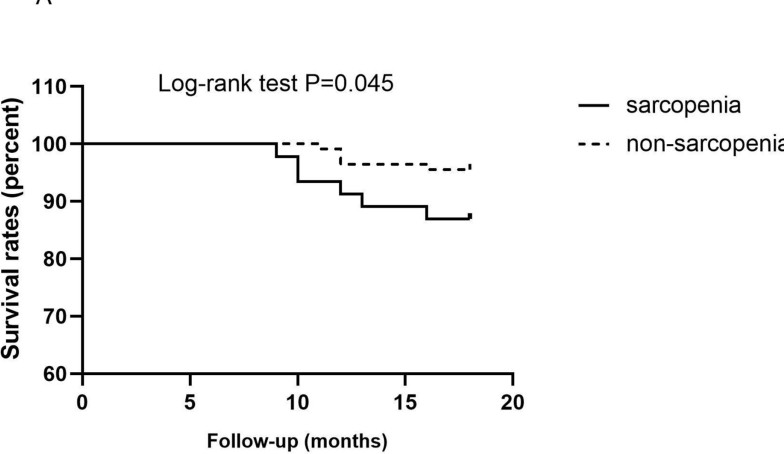

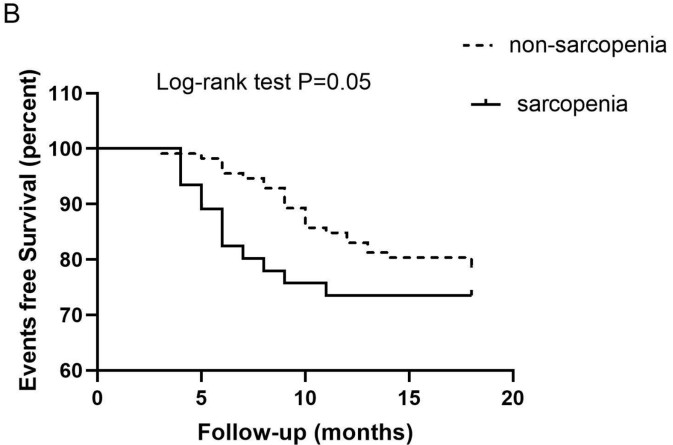

**Figure 3 Kaplan–Meier analysis for the probability of survival (A) and cardiovascular events (B) according to the presence of sarcopenia in patients on maintenance hemodialysis.**

## DISCUSSION

In this prospective cohort study of patients undergoing MHD, our principal findings were that sarcopenia is associated with cardiac structure and function, as assessed by echocardiography. Compared with non-sarcopenic participants, patients with sarcopenia had a smaller RV size, IVST, and LVPW. Interestingly, we found a strong positive association between SMM and LVM. Additionally, logistic regression analysis showed that patients with sarcopenia may have a higher risk of developing LVDD and LVH. The follow-up results showed that sarcopenia was a significant predictor of all-cause mortality and CVE in patients undergoing hemodialysis.

Sarcopenia is used to describe the loss of muscle mass and function due to aging, development of chronic diseases, physical inactivity, and inappropriate nutrition (*Cruz-Jentoft & Sayer, 2019*). Recently, the effect of sarcopenia on long-term clinical outcomes

**Table 5 Multivariate cox regression analyses for all-cause mortality and cardiovascular events.**

| Variable | All-cause mortality | | | Cardiovascular events | | |
|---|---|---|---|---|---|---|
| | Hazard ratio | 95% CI | P-value | Hazard ratio | 95% CI | P-value |
| Age, y | 1.10 | [1.02–1.18] | 0.019 | 1.09 | [1.00–1.20] | 0.061 |
| Prevalent CVD | 1.95 | [0.41–9.38] | 0.405 | 4.30 | [0.50–36.87] | 0.183 |
| Albumin, g/L | 1.20 | [0.95–1.53] | 0.134 | 1.27 | [0.93–1.72] | 0.128 |
| SMI, kg/m² | 3.41 | [1.01–11.57] | 0.049 | 3.54 | [0.81–15.58] | 0.094 |
| HGS, kg | 0.88 | [0.78–0.99] | 0.037 | 0.86 | [0.74–1.01] | 0.071 |
| Sarcopenia | 6.59 | [1.08–39.91] | 0.041 | 10.96 | [1.14–105.10] | 0.038 |
| BMI, kg/m² | 1.01 | [0.78–1.30] | 0.965 | 1.05 | [0.75–1.47] | 0.775 |
| LVMI, g/m² | 0.99 | [0.97–1.01] | 0.479 | 1.00 | [0.97–1.02] | 0.786 |

Notes:
Multivariate Cox regression analysis was performed after adjusting for age, sex, BMI, Kt/V, albumin, presence of diabetes, hypertension, CRP, low skeletal muscle index, handgrip strength, and previous history of cardiovascular disease and LVMI.
CI, confidence interval; SMI, skeletal muscle index; HGS, handgrip strength; BMI, body mass index; CVD, cardiovascular disease; CRP, C-reactive protein; LVMI, left ventricular mass index; Kt/V, a measure of dialysis adequacy; K, dialyzer clearance of urea; t, dialysis time; V, volume of urea distribution.

has become clearer; sarcopenia has been recognized as an important prognostic marker in various populations (*Tandon et al., 2021*; *Sepulveda-Loyola et al., 2020*; *Yin et al., 2019*).

Sarcopenia was revealed as an independent risk factor for LVDD in a large population ($n$ = 31,258, aged ≥20 years) who underwent health examinations (*Yoo et al., 2021*). A retrospective study also demonstrated the association between sarcopenia and LV diastolic function in patients with type 2 diabetes. LVDD is prevalent in patients undergoing hemodialysis and has been proven to be a risk factor for CVE (*De Lima et al., 2022*). Herein, the prevalence of LVDD was 64.6%, which is close to that observed in other studies (*Barberato et al., 2010*; *de Bie et al., 2012*). Adverse effects on left ventricular diastolic function in patients on MHD were independently associated with a higher probability of death and CVE (*Han et al., 2015*; *De Lima et al., 2022*). In our study, we found a significant difference in the presence of LVDD between sarcopenic and non-sarcopenic patients. Multivariate logistic regression showed that sarcopenia was associated with LVDD after adjusting for confounding factors, with a 4.91-fold increased risk in LVDD.

CVD is the leading cause of death in patients undergoing MHD, and LVH is an important predictor of cardiovascular mortality and morbidity in patients undergoing dialysis (*McCullough et al., 2016*). The findings of this study indicate that sarcopenia is associated with an increased risk of LVH, suggesting that SMM plays an independent role in the pathogenesis of LV remodeling. The results in this study are consistent with those in a large cohort study of Korean adults ($n$ = 67,106), which showed an increased prevalence of both LVDD and LVH in the sarcopenia group (*Ko et al., 2018*).

These findings are not surprising. Evidence suggests that sarcopenia may not only significantly reduce the quality of life of patients undergoing MHD, but that it is also associated with CVD risk factors such as age, sedentary lifestyle, obesity, insulin resistance, and metabolic syndrome, which increase the risk of CVD complications and death (*Rysz et al., 2020*). Herein, we found that approximately one-third (33.1%) of patients undergoing

HD had sarcopenia; the presence of sarcopenia and low muscle mass caused a 6.58- and 3.41-fold increase in the risk of all-cause mortality, respectively. An increased HGS caused a 0.88-fold drop in the risk of all-cause mortality. The presence of sarcopenia was associated with CVE and might have caused a 10.96-fold increase in the incidence of CVE.

The mechanisms underlying the association between sarcopenia and CVE are not yet fully understood. Obesity, dyslipidemia, inflammatory response, and insulin resistance due to reduced or loss of exercise ability caused by sarcopenia promote the occurrence of CVD (*Barbalho et al., 2020*). In animal model experiments, Akt (protein kinase B)-mediated skeletal muscle secretion of substances called cardiac protective factors have protective endocrine effects, such as stimulating the growth of muscle and mast cells, accelerating fat oxidation, enhancing insulin sensitivity, and mediating anti-inflammatory effects, which lead to a reduction in myocardial damage (*Araki et al., 2012*). Pathological changes in skeletal muscle in patients with sarcopenia may reduce the protective effects of cardiac protective factors (*Adams et al., 2022*).

Interestingly, we found a strong positive association between SMM and LVM, and participants with sarcopenia had a smaller RV cavity size, IVST, and LVPW. These results may confirm the existence of the cardio-sarcopenia syndrome, wherein sarcopenia affects both the skeletal muscle and myocardial systems (*Keng et al., 2019*). High LVM is traditionally viewed as a clinically unfavorable phenomenon, and LVH in hypertensive pathology is associated with greater cardiovascular risk and a poor prognosis (*Reinier et al., 2011*). Conversely, the potential role of LVH as a secondary compensatory mechanism for increasing cardiac work should also be emphasized, as seen in physiological LVH in athletes (*Nakamura & Sadoshima, 2018*).

To the best of our knowledge, this is the first study to focus on the correlation between sarcopenia and cardiac structural and functional parameters in patients on MHD, which indicates a predictive role for sarcopenia in LVH and LVDD. We highlighted the presence of sarcopenia as a potential risk marker for changes in cardiac structure and function, which further affects the prognosis of patients on MHD. We believe that our findings may contribute to a better knowledge of sarcopenia and its possible adverse effects in MHD patients. It is hoped that necessary clinical nutrition or rehabilitation interventions can be taken in the early stage of the disease in time to improve the prognosis of these patients.

Our study has several limitations. First, the single-center nature of our study limited the number of patients that could be included. Second, due to the inclusion criteria of this study, we did not include immobile or bedridden patients who might be eligible for the diagnosis of sarcopenia, leading to an underestimation of the prevalence of this condition at our center. Third, the follow-up period was relatively short, lasting only 18 months.

## CONCLUSION

Sarcopenia was an independent risk factor for LVDD and LV remodeling in MHD. Patients with sarcopenia on MHD have high rates of CVE and mortality. Although larger studies are required to confirm our findings, our results allow us to conclude that sarcopenia can be a useful therapeutic target for cardiovascular risk reduction in these patients.

## ACKNOWLEDGEMENTS

The authors thank all patients who participated in this study and the HD treatment staff at the Zhongda Hospital for their contributions to this study.

### Funding

Our work was supported by the grants from the National Natural Science Foundation of China (82000648), the Natural Science Foundation of Jiangsu Province (BK20200363), the Outstanding Youth Cultivation Foundation of Southeast University (2021ZDYYYQPY07), and the Innovative and Entrepreneurial Talent (Doctor) of Jiangsu Province. The funders had no role in study design, data collection and analysis, decision to publish, or preparation of the manuscript.

### Grant Disclosures

The following grant information was disclosed by the authors:
National Natural Science Foundation of China: 82000648.
Natural Science Foundation of Jiangsu Province: BK20200363.
Outstanding Youth Cultivation Foundation of Southeast University: 2021ZDYYYQPY07.
Innovative and Entrepreneurial Talent (Doctor) of Jiangsu Province.

### Competing Interests

The authors declare that they have no competing interests.

### Author Contributions

- Mengyan Zhang conceived and designed the experiments, performed the experiments, analyzed the data, authored or reviewed drafts of the article, and approved the final draft.
- Liuping Zhang conceived and designed the experiments, performed the experiments, analyzed the data, authored or reviewed drafts of the article, and approved the final draft.
- Yezi Hu performed the experiments, analyzed the data, prepared figures and/or tables, and approved the final draft.
- Ying Wang performed the experiments, analyzed the data, prepared figures and/or tables, and approved the final draft.
- Shengchun Xu performed the experiments, authored or reviewed drafts of the article, and approved the final draft.
- Xiaotong Xie analyzed the data, authored or reviewed drafts of the article, and approved the final draft.
- Tian Xu analyzed the data, authored or reviewed drafts of the article, and approved the final draft.
- Zuolin Li analyzed the data, authored or reviewed drafts of the article, and approved the final draft.
- Hui Jin analyzed the data, authored or reviewed drafts of the article, and approved the final draft.

![PeerJ](PeerJ logo)

- Hong Liu analyzed the data, authored or reviewed drafts of the article, and approved the final draft.

## Human Ethics

The following information was supplied relating to ethical approvals (*i.e.*, approving body and any reference numbers):

The study was approved by the Independent Ethics Committee (IEC) for Clinical Research of Zhongda Hospital, affiliated with Southeast University (batch number: 2021ZDSYLL230-P01) and was conducted in compliance with the tenets of the Declaration of Helsinki. All the patients provided written informed consent for participation in the study and data processing.

## Data Availability

The raw data is available in the Supplemental File.

## Supplemental Information

Supplemental information for this article can be found online at http://dx.doi.org/10.7717/peerj.14429#supplemental-information.

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
