# Peer review of "Sarcopenia and echocardiographic parameters for prediction of cardiovascular events and mortality in patients undergoing maintenance hemodialysis"

_PeerJ, doi:10.7717/peerj.14429_

## Round 0.1 · original submission · Major Revisions

Dear authors,

The manuscript needs major revisions to improve.

Please reply point by point to the reviewers' comments. Follow their suggestions carefully.

·

Basic reporting

The study represents a valid exploration of the relationship between sarcopenia and heart disease. The background is sufficient and the approach used is correct as a cohort study. Valid are the instruments used and the sample examined.

Experimental design

The correlational objective of the study is clear but a research hypothesis should be specified. The experimental design based on the "cohort study" model also examined the follow up of the subjects examined. Applied an adequate correlational method and tools whose validation is appropriate to the type of study.

Validity of the findings

The results are sufficient to demonstrate the validity of the study and the limitations of the present study are also well identified. Future applications and studies could be specified.

Additional comments

In the references it is necessary to standardize the font size of the Journal cited. Also uniform if pointed or in full. for example: "CHINESE MEDICAL JOURNAL "or" Chin Med J. "

·

Basic reporting

1. „Background“ section of the „abstract“ should be rewritten. Either you should change the position of first sentences, or simply lose the first two, and start straightforward with „We studied the association between sarcopenia and cardiac structure and function in patients …“.
Introduction is in general too short. It merely touches some key points. Even if they were mentioned, authors honestly admit that “Recent studies have shown…” what their study shows…
2. A sentence in ln 61/62 is awkwardly said.
3. Ln 62/63 – do you mean compared to or compared with? It is not fully clear
4. Please, explain (be more specific) on decreased activity in ln 64.
5. Authors used <ln 67 to lure us into deceptive topic of sarcopenia yet to reveal that “Recent studies have shown that changes in skeletal muscle mass (SMM) are closely related to CVD, and sarcopenia can increase the incidence of cardiovascular events (CVE)“ – why? OR what is the difference between these results and those of Lena et al. or that of Veronese et al. ?
6. How can anyone explain sudden glimpse towards Left ventricular hypertrophy (LVH) and left ventricular diastolic dysfunction (LVDD)?
7. Is the reference to Veronese et al. really necessary at this point (ln 74)? It was cited ffew rows above, and at this point you have Kim et al.,, as well
8. I had great hopes for” cardio-muscular axis”, unfortunately, this idea has been merely mentioned.

Experimental design

n/a

Validity of the findings

N/A

Reviewer 3 ·

Basic reporting

I was full of expectance when I first glanced this paper, so my disappointment was even greater as I was reaching my way through. Final paragraph of the “introduction” is imaginary “conclusion” of a hypothetical paper I would like to read. In this form, it is just a poorly organized pile of misleading facts. Paper lacks straightforward hypothesis, so I yet in “results” found out what was really the objective of whole narrative. Funny, I was almost misled by all those “smart pants” facts. Luckily, the authors themselves admit that “Recent studies have shown that changes in skeletal muscle mass (SMM) are closely related to CVD, and sarcopenia can increase the incidence of cardiovascular events (CVE)“.

Experimental design

not assessed

Validity of the findings

basically, no value since this is a well studied topic. You should start with cardio-muscular axis and run whole paper backwards.

·

Basic reporting

- The greatest concern is the definition of sarcopenia used in the manuscript versus the published AWGS. Low ASM + low muscle strength OR Low physical performance. The authors did not state how they used BIA to determine ASM and how grip versus gait speed were used. The confusion is demonstrated in the interpretation in the Discussion line 224 where it is stated sarcopenia and low muscle mass caused a….The definition of sarcopenia inherently includes low muscle mass. It appears that only grip strength or gait speed were used to define, but overall the lack of clarity limits the readers ability to fully appreciate the study and its findings.
- The authors generally do a good job of using clear and unambiguous professional English. Some areas of improvement include lines 55, 61, 245, 246, 247.
- The article presented the figures and tables as supplemental information however there was some mislabeling of tables versus figures. It made it cumbersome while referring to figures and tables.
- Text in figures 1-3 could be improved by increasing font size and/or increasing clarity of text as some is blurry.
- Tables appear to be copy/paste of image that are of lower quality.
- Paragraph and line spacing in introduction needs to be corrected, lines 61/62, 66/67.
- Sufficient background information was present, and the results were related to the aims. A clearer hypothesis would be beneficial.
- What is the difference between Lai et al., 2019a vs Lai et al., 2019b?
- Authors use numbered references in discussion line 209. Fix formatting and describe if the values are in CKD population or other.

Experimental design

- The research fits well within the aims and scope of the journal as it involves original primary research in the health and medical sciences.
- The article clearly establishes the gap (i.e. relationship between sarcopenia and cardiac structure and function) and how they are filling that gap.
- Tests and measures need to be described with greater detail
o Physical disability line 89
o BIA was performed after dialysis, but what about physical function measures?
o Define adequate dialysis
o Grip strength position stated both seated and standing, which one?
o Define sarcopenia criteria, did the participant need to be below both cutoffs for gait speed and grip strength?
- The investigators were careful avoid inaccurate results by performing BIA after dialysis. BIA is a measure of skeletal muscle but other methods such as DXA provide much more accurate data.
- Methods are clearly written, and sufficient detail is provided.

Validity of the findings

- The investigators did not present the raw data. The supplemental figures would benefit from more thorough descriptions to be more useful to readers.
- The statistics used in the paper seem appropriate for the analysis’ being completed.
- Table 1
o parcel out male/female grip strength values
o if <1m/s is cutoff for sarcopenia as listed currently or assumed currently, why is the non-sarco gait speed mean 0.93m/s?
- The conclusion fits well with the aims originally stated.
- Discussion line 198- consider deleting as this could be used in the intro to build the story, but not needed here.

---

## Round 0.2 · accepted · Accept

Dear authors,

Congratulations! The manuscript is now ready for publication.

Best wishes!

·

Basic reporting

I find this paper much improved and ready to be published. Presentation and narration styles have all been enhanced, and clarity of style has been achieved

Experimental design

see above

Validity of the findings

see above

Reviewer 3 ·

Basic reporting

This paper seems fine now, so it is ok so publish. All questions have been addressed as properly.

Experimental design

N/A

Validity of the findings

n/a

Additional comments

n/a